# Recent Advances in Asymmetric Synthesis of Pyrrolidine-Based Organocatalysts and Their Application: A 15-Year Update

**DOI:** 10.3390/molecules28052234

**Published:** 2023-02-27

**Authors:** Arianna Quintavalla, Davide Carboni, Marco Lombardo

**Affiliations:** 1Alma Mater Studiorum—Department of Chemistry “G. Ciamician”, University of Bologna, Via Selmi 2, 40126 Bologna, Italy; 2Center for Chemical Catalysis—C3, Alma Mater Studiorum—University of Bologna, Via Selmi 2, 40126 Bologna, Italy

**Keywords:** asymmetric organocatalysis, proline, substituted pyrrolidines, synthetic methods

## Abstract

In 1971, chemists from Hoffmann-La Roche and Schering AG independently discovered a new asymmetric intramolecular aldol reaction catalyzed by the natural amino acid proline, a transformation now known as the Hajos–Parrish–Eder–Sauer–Wiechert reaction. These remarkable results remained forgotten until List and Barbas reported in 2000 that L-proline was also able to catalyze intermolecular aldol reactions with non-negligible enantioselectivities. In the same year, MacMillan reported on asymmetric Diels–Alder cycloadditions which were efficiently catalyzed by imidazolidinones deriving from natural amino acids. These two seminal reports marked the birth of modern asymmetric organocatalysis. A further important breakthrough in this field happened in 2005, when Jørgensen and Hayashi independently proposed the use of diarylprolinol silyl ethers for the asymmetric functionalization of aldehydes. During the last 20 years, asymmetric organocatalysis has emerged as a very powerful tool for the facile construction of complex molecular architectures. Along the way, a deeper knowledge of organocatalytic reaction mechanisms has been acquired, allowing for the fine-tuning of the structures of privileged catalysts or proposing completely new molecular entities that are able to efficiently catalyze these transformations. This review highlights the most recent advances in the asymmetric synthesis of organocatalysts deriving from or related to proline, starting from 2008.

## 1. Introduction

Substituted chiral pyrrolidines represent one of the most common heterocyclic structural motifs that are present in biologically active natural and synthetic compounds [1,2]. Meanwhile, this scaffold also plays a crucial role as a building block in organic synthesis and it characterizes the structure of many ligands [3,4]. Furthermore, since the advent of organocatalysis, chiral pyrrolidines have assumed a leading position as organocatalysts [5,6], as they are able to efficiently promote several different transformations in an enantioselective and environmentally friendly way, avoiding the use of metals. In this context, an important challenge is represented by the design and synthesis of structurally innovative organocatalysts. Starting from the first employed entities, such as natural proline and derivatives [7,8], MacMillan’s imidazolidinones [9], and Jørgensen and Hayashi’s diarylprolinol silyl ethers [10,11] (Figure 1a), a huge number of diverse pyrrolidine-based chiral organocatalysts have been proposed. The peculiar structure and substitution pattern of these organocatalysts determine the activation mode of the catalyst, leading to an enantioselective transformation (Figure 1b–e) [12,13].

In fact, chiral pyrrolidines, characterized by a hydrogen bonding donor in the C-2-side chain (proline, prolinamides, prolinols, triazole, etc.; Figure 1c), usually act by covalently binding a substrate and coordinating the second one through hydrogen bonds. Conversely, chiral pyrrolidines, lacking hydrogen bonding donors at C-2 and characterized by high steric hindrance or structural rigidity (Figure 1d,e), act by shielding one face of the substrate and driving the attack onto the other face. Based on the knowledge of their mode of action, the structures of pyrrolidine-derived organocatalysts have been extensively modified with the aim of optimizing the efficiency and selectivity of the catalysts and adapting them to the use of increasingly complex or less reactive substrates.

The huge interest in this structural motif has led to great efforts in developing novel and efficient synthetic strategies for the asymmetric construction of substituted chiral pyrrolidines. Some reviews surveyed an aspect of the synthetic protocols according to a peculiar focus, usually related to a specific chemical transformation: asymmetric syntheses of pyrrolidines exploiting organocatalysis [14], asymmetric [3 + 2]-cycloadditions involving azomethine ylides [15], catalytic asymmetric 1,3-dipolar cycloadditions [16], aryl-sulfinamides in the synthesis of *N*-heterocycles [17], synthesis of *N*-phosphorylated pyrrolidines [18], annulation strategies for the construction of multisubstituted pyrrolidines as natural products [19], or palladium-catalyzed alkene aminoarylation reactions [20]. Conversely, in this review, we focus on the advances achieved in the last 15 years (between 2008 and 2022) in the asymmetric synthesis of novel chiral pyrrolidine-based organocatalysts, including all of the proposed synthetic approaches. The catalysts developed in this period have been divided and described in three chapters (proline-related, prolinol-related, and diarylprolinol-related organocatalysts) according to their key structural features and, consequently, their activation mode.

We included the catalysts in which the structural variations involve the interaction with the substrates and, therefore, the transition state formation. The supported/immobilized/heterogeneous catalysts were considered beyond the scope of this survey of the literature, because the structural modifications are aimed at the catalyst’s recovery and reuse. Readers are kindly addressed to specific reviews covering the topic of pyrrolidine-based recyclable organocatalysts [21,22,23,24,25,26].

## 2. Proline-Related Organocatalysts

Proline can be considered the simplest aldolase in terms of structure, and since the seminal reports by List and Barbas on the direct asymmetric aldol reaction [27,28], proline has been successfully employed in a variety of asymmetric transformations [7]. Proline has assumed a leading role as a chiral privileged organocatalyst due to the presence of (i) a secondary amine involved in a five-membered ring, able to promptly form structurally defined enamines by reacting with enolizable carbonyl compounds, and (ii) a carboxylic acid which is able to coordinate a reacting electrophilic partner through hydrogen bonding interactions and to drive it specifically towards one of the two diastereotopic faces of the transient enamine [29]. Despite the great advantages offered by proline as an organocatalyst, some drawbacks limit its general application in synthetic transformations. First, proline is mostly insoluble in organic solvents; therefore, high catalyst loadings were commonly used to obtain acceptable reactivities. Moreover, the achieved enantioselectivities were moderate (<90% *ee*) and it was not usable in water, nowadays attracting great interest as an abundant, safe, and environmentally friendly reaction medium. On these bases, a huge number of structurally different organocatalysts were designed and synthesized to overcome these limitations, mainly introducing different groups on the C2-position of the pyrrolidine ring which were still able to act as efficient hydrogen-bond donors.

### 2.1. Prolinamides

Among the huge number of structurally different organocatalysts developed to overcome the limitations of proline, prolinamides enjoyed great success. In fact, they maintain the typical mode of action of proline coupled with improved properties due to the amidic sidechain. On the basis of their structure, these organocatalysts can be divided into “simple” amides and those containing additional functional groups which are capable of participating in the substrates coordination. Chiral prolinamides were exploited mainly to promote the asymmetric direct **aldol reaction**; however, some examples of the **conjugate addition** of carbonyl compounds to electron-poor π-systems were also reported. Since the synthesis of prolinamides generally relies on well-established synthetic procedures developed for the synthesis of peptides, this chapter will mainly focus on the structure of the newly proposed organocatalysts and on the peculiar activation modes of the reaction partners.

In 2010, Carter and coworkers proposed the new ester-containing proline aryl sulfonamide **1** (Figure 1), which is able to catalyze the conjugate addition of racemic α,α-disubstituted aldehydes to acyclic unsaturated ketones, leading to cyclohexenones containing two adjacent stereocenters and including an all-carbon-quaternary carbon in high enantio- and diastereo-selectivity [29,30]. In this multicomponent coupling, benzylamine was crucial, presumably forming the enamine of the aldehyde which reacts with the chiral iminium ion of the ketone (Figure 1). The lipophilic substituent on the aryl sulfonamide significantly improved the catalystsolubility in apolar solvents and its reactivity compared to proline.

In 2012, Sirit et al. investigated the enantioselective Michael addition of aldehydes to nitrostyrenes in apolar solvents mediated by two prolinamides which were fully substituted on the amide moiety (Figure 2), thus lacking the hydrogen-bond donor group typical of proline [31]. The use of a co-catalyst containing an acidic hydroxyl group was mandatory to achieve good performance, with (*S*)-1,1′-bi-2-naphthol providing the best results as a matched pair. The authors suggested a transition state model organized by a network of hydrogen bonding interactions (Figure 2), in which the amide acts as bulky substituent and the better enantioselectivity of catalyst **3** with respect to **2** could derive from the additional stereocenter.

The same asymmetric conjugate addition was investigated in 2012 by Kelleher’s research group, which proposed a series of 4-hydroxyprolinamides (Figure 2b) to examine the impact of the 4-hydroxy and the 2-methyl groups on the enantioselectivity of the prolinamide organocatalysts [32]. The synthetic route started with the α-methylation of the fully protected *trans*-4-hydroxyproline, followed by the diastereoisomers separation, the usual steps for the amide synthesis, and the 4-hydroxy-deprotection (Figure 2a). The coupling with *N*,α-dimethylbenzylamine provided the *N*-methylated prolinamides in low yields, due to the hindered nature of both of the coupling partners. The final Boc-removal proved to be problematic due to the high aqueous solubility of the products. For comparison, the analogous simple 4-hydroxy NH and *N*-methyl prolinamides were prepared in a similar manner. The application of all of the catalysts in the model conjugate addition showed that: (i) the *cis* relative position of the 4-OH was detrimental in the *N*-methyl amide, (ii) the removal of the α-methyl group provided improved results, (iii) the simplest hydroxyprolinamides **6c** and **6d** gave excellent performance, and (iv) replacing the *N*-1-phenylethyl side-chain to either a more, or less, sterically hindered moiety was detrimental to the stereoselectivity. DFT calculations suggested that the key role was played by the structure of the amide sidechain affecting the relative stability of the *E*-enamines derived from each catalyst. Small changes in the substrates structure can lead to profound differences in the enamine’s stabilities and transition states.

Lin and Wei, in 2014, exploited the reactivity of two new prolinamides characterized by bulky achiral substituents on the amide function (Figure 3) [33]. The most efficient Michael addition was achieved with the most hindered catalyst (*S*)-*N*-tritylpyrrolidine-2-carboxamide **7b** (10 mol%) at −20 °C, which provided good yields and enantiocontrol but moderate *dr* with some substrates.

Afterwards (2019), Ramapanicker and co-workers prepared five D-prolinamides from the corresponding 2-(trifluoromethylsulfonamidoalkyl)pyrrolidines (Figure 3), to be used in the Michael addition of aldehydes to β-nitroalkenes at room temperature without additives [34]. The authors envisaged that the trifluoromethanesulfonamide (−NHTf) group could offer useful catalytic performance, acting as an H-bond donor instead of the carboxylic or phosphonic acids present in the peptidic catalysts which were previously proposed by Wennemers [35] and Lecouvey [36]. The synthetic strategy started with the preparation of the NHTf-substituted pyrrolidines in four high yielding steps from the corresponding aldehydes (reductive amination, hydrogenolysis, sulfonamide synthesis, and *N-*Boc deprotection; Figure 3). The following coupling with *N-*Boc-D-proline and deprotection furnished the catalysts. The superior results obtained in the nitro-Michael reaction with catalyst **8c** suggested that an optimal distance between the secondary amine and the hydrogen bond donor (NHTf) is needed. Good yields and stereoselectivities were observed; however, an excess of aldehyde, 10 mol% of catalyst, and at least two days were required.

Regarding the asymmetric direct aldol reaction, in 2008, Nájera et al. reported on the preparation of two new prolinamides derived from (1*S*,2*R*)-*cis*-1-aminoindan-2-ol and (*R*)-1-aminoindane, and on the corresponding prolinethioamides, using Lawesson’s reagent (Figure 4) [37]. Their reactivity was evaluated in enantioselective solvent-free inter- and intramolecular aldol reactions. The best results were achieved with thioamide **9d**, which showed good performance in low loading (5 mol%) and with only one equivalent of ketone. Further, **9d** proved to be a superior catalyst to L-Pro and the corresponding prolinamides for the solvent-free tested reactions. Moreover, it was easily recovered and reused by a simple acid/base work-up.

The same research group later proposed two pyrimidine-derived prolinamides containing (*R,R*)- or (*S,S*)-*trans*-cyclohexane-1,2-diamine as bifunctional organocatalysts (Figure 5) to be applied in the same solvent-free aldol processes [38]. DFT calculations suggested that the aldehyde is coordinated by two NH groups through H-bonding, explaining the preferential formation of the *anti*-isomer with both the diastereomeric catalysts (proline stereocenter governs the stereoselectivity) and confirming the beneficial role of the bifunctional system. Both the inter- and intra-molecular aldol reactions were scaled up to 1 g, and the organocatalysts were recovered (extractive work-up) and reused without a significant loss in activity.

In 2009, Chen and co-workers synthesized a family of pyrrolidinyl-camphor-containing bifunctional organocatalysts, deriving from proline or 4-hydroxyproline, by means of simple synthetic steps (Figure 4) [39]. They were designed to synergistically activate both the nucleophilic and electrophilic reacting partners via enamine and hydrogen bonding. Moreover, the rigid bicyclic camphor moiety can act as a stereocontrolling element. The stereo and electronic properties of the catalysts were finely tuned through structural modifications and the obtained amphiphilic organocatalysts were tested in the asymmetric aldol reaction either in organic solvents or in the presence of water. Excellent yield and stereocontrol were achieved with thiourea **14c** in the model reaction in water, although 20 mol% of both catalyst and additive (dodecylbenzenesulfonic acid acting as both acid promoter and surfactant) and 36 h were required.

A series of prolinamides, characterized by different carbocyclic rings or variably substituted aromatics (Figure 6), was synthesized in 2011 by Fu et al., aiming to establish a simple, inexpensive, and efficient route to enantiomerically enrich aldol products for industrial applications [40]. The simplest and least expensive catalyst, **17a**, showed the best performance working in *m*-xylene at −20 °C. This catalyst was successfully employed in large-scale reactions (150 mmol of aromatic aldehyde), easily recovered by acid treatment, and reused for five cycles without a significant drop of enantioselectivity.

In the same year, Moorthy and coworkers applied a series of new prolinamides in the enantioselective organocatalytic Biginelli reaction [41]. The catalysts, containing one or two H-bonding donors, were easily synthesized starting from L-proline or *trans*-4-hydroxy L-proline (**7b,c**, **18a–h**–**20**, Figure 5a), and the *cis*-4-tosylamide derivatives (**18i,j**) were obtained by exploiting the usual strategy for OH-replacement with NH_2_ (Figure 5b). The sterically hindered *N*-arylprolinamide **18j**, having chirality reinforced by an additional stereocenter and a very acidic hydrogen bond donor at C-4, performed well with uniformly high enantiocontrol (94–99%) in the cyclocondensation of aliphatic as well as aromatic aldehydes with urea and ethyl acetoacetate, and in the presence of pentafluorobenzoic acid and tritylammonium trifluoroacetate as additives, which increased the yields and *ee*s.

A different approach, based on plant-derived chiral β-aminoalcohols, to construct bifunctional prolinamide-based organocatalysts was proposed in 2011 by Zlotin et al., which conjugated (*S*)-proline with stereoisomeric 2-hydroxy-3-aminopinanes (Figure 7) [42]. The catalytic properties were studied in the asymmetric aldol reaction in water, as it is an environmentally friendly medium. The authors assumed that catalyst **21b**, owing to its *syn*-orientated NH– and OH– groups, provided better results thanks to the more efficient hydrogen bonding network with the aldehyde in the transition state of the addition.

The aldol reaction of isatin with acetone was selected by Lennon and coworkers in 2011 to evaluate the reactivity and selectivity of a family of simple prolinamides, among which were *N*-pyridyl and *N*-quinolinyl derivatives (Figure 6) [43]. The authors hypothesized that a combination of hydrogen bonding and π–π stacking of the aromatic rings of the catalysts with isatin were responsible for the observed catalytic performance. The nitrogen position in the aromatic ring had a significant effect on the enantioselectivity, with the 3- and 4-aminopyridinyl catalysts **22c** and **22f**, and the 3-aminoquinolinyl catalyst **22i**, yielding the best results. The presence of a second stereogenic center in the organocatalyst led to inferior stereocontrol. The amide N–H played a crucial role and an enhanced N–H acidity led to interesting results (catalysts **17e** and **24a**, the last containing a sulfonamide). Conversely, the binding to the N–H of isatin may not have played a significant role.

In the same year, Nakamura’s group investigated the enantioselective cross-aldol reaction between acetone and trihalomethyl ketones or imines promoted by *N*-(8-quinolinesulfonyl)prolinamide **24f** and found it to be the best organocatalyst (Figure 7) [44]. The MO calculations suggested that the hydrogen bonding between the sulfonimide proton and the 8-quinolyl nitrogen plays a remarkable role in determining the reaction enantioselectivity.

In 2012, Ventura’s group described the synthesis of a series of new pyrrolidine-based organocatalysts derived from tartaric and glyceric acids (Figure 8) [45]. The authors supposed that the addition of extra stereocenters and suitably positioned functional groups can contribute to create new effective organocatalysts. The synthesis of catalysts **25a–d** was accomplished in eight and nine steps, starting from dimethoxy-2,3-dimethyl-1,4-dioxane-5,6-dicarboxylate **I-9**, obtained in turn from L- or D-tartaric acid (Figure 8a). The *cis*-catalyst **25e** was prepared with some difficulties from dithioester **I-12** (Figure 8b). Organocatalyst **25f** was readily obtained from glyceric acid methyl ester (Figure 8c) and organocatalyst **26g** was prepared from bicyclic product **I-17** (Figure 8d). The application of these catalysts to an asymmetric aldol reaction led to modest results. The catalysts were not generally applicable, probably because of the high conformational rigidity conferred by the dioxane ring.

A structural variation of the sulfonamide-based organocatalysts already proposed by Carter [30] and Lennon [43] was developed by Wan and Hao in 2014, who proposed *N*-prolyl sulfinamides containing two stereogenic centers (Figure 8) [46]. Four organocatalysts (neutral sulfonamide **27a**, its corresponding enantiomeric TFA salts **27b**, **27c**, and diastereomeric salt **27d**) were applied in the asymmetric aldol reaction under the solvent-free condition, it being a more efficient and environmentally friendly system. Good performance was achieved with **27b** at 0 °C, although some days were required to reach high conversion. The reactivity of neutral **27a** was inferior, whereas a much poorer enantioselectivity and reactivity were observed with diastereomeric catalyst **27d**, as it presented a reversed asymmetric induction, confirming that the enantioswitching is dominated by the configuration of the prolyl unit. (*S*)-*N*-(methylsulfonyl) pyrrolidine-2-carboxamide promoted the same reaction with worse results, supporting the profitable role of the additional stereocenter that is present in these sulfinamides.

In 2014, the phthalimido-prolinamide **28** (Figure 9) was developed by Kumar et al. to promote the enantioselective direct aldol reaction of aromatic aldehydes with ketones under solvent- and additive-free conditions [47]. A small amount of water (5 mol%) improved the reaction rate and, slightly, the enantioselectivity. The authors supposed that water contributes to creating a more compact transition state, leading to higher stereoselectivity (Figure 9).

New bifunctional organocatalysts based on a nucleoside and proline, named AZT-prolinamides (Figure 10), were synthesized by Chandrasekhar et al. in 2014 to mediate the same enantioselective aldol reaction in additive-free water [48]. The results obtained with catalyst **29b** were slightly inferior to those obtained for catalyst **29a**, with both used at 15 mol% loading. The authors suggested that the nucleoside moiety acts as a steric controller and contributes to hydrogen bond stabilization along with the amide group. The prolinamide **29a** provides an additional hydrogen bond due to the presence of the hydroxy group (A); conversely, the silylated prolinamide **29b** exerts extensive steric coverage (B).

Some of the 4-hydroxy-prolinamides (Figure 11) synthesized by Kelleher et al. [32] in 2012 were applied by Singh in 2015 in the asymmetric aldol reaction using low catalyst loading (10 mol%) under solvent-free conditions, according to the green chemistry principles [49]. The trans-4-Hydroxy-(*S*)-prolinamide **6d**, containing (*S*)-1-phenylethylamine, proved to be the best catalyst. The (*S*)-1-phenylethylamine and the additional 4-hydroxyl group in the trans position both positively influenced the enantioselectivity of the predominant anti-aldol product, with the 4-OH probably forming a network of intermolecular hydrogen bonds (Figure 11). Catalyst **6d** can be reused for up to five cycles for the asymmetric aldol reaction between cyclohexanone (1.2 mmol) and p-nitrobenzaldehyde (0.4 mmol) using acetic acid (10 mol%) as the additive at 15 °C, obtaining a slightly lower *ee* but an improved *dr*.

In 2016, the same prolinamides, with or without the 4-hydroxy group, were also applied by Singh as organocatalysts in the asymmetric aldol reaction of isatins with acetone [50]. With the best catalyst, **6d** (Figure 11), the reaction proceeded at −35 °C with acetone as a co-solvent under additive-free conditions, providing the aldol products in high yields (up to 99%) and with moderate enantioselectivities (up to 80%).

The asymmetric aldol reaction between acetone and aromatic aldehydes was selected to evaluate the performance of the novel bifunctional organocatalysts proposed by de Parrodi and coworkers in 2015 [51]. The homochiral L-prolinamido-sulfonamides **30a–h** (Figure 12) were easily synthesized from enantiomerically pure (*R,R*)-11,12-diamino-9,10-dihydro-9,10-ethanoanthracene as a 1,2-diamine structural motif to be derivatized as both prolinamide and sulfonamide. The best results were achieved with catalyst **30d**, bearing a strong electron-withdrawing substituent; however, the reaction scope was limited. To confirm the role of the sulfonamide-NH, the model reaction was carried out with the corresponding -NMe catalyst, leading to worse yield and stereocontrol. The diastereoisomeric organocatalyst obtained from the enantiomeric 1,2-diamine also provided worse *ee*. The calculated transition state structure evidenced an intramolecular hydrogen bond between the sulfonamide NH and the prolinamide C=O which may lock the proline conformation, affecting the selectivity.

### 2.2. Peptides

Peptidic organocatalysts have a modular nature which allows for easily change both their structure and properties while maintaining the profitable hydrogen bonding interactions that are typical of proline. As already mentioned, for prolinamides, the synthesis of peptides is accomplished through well-established synthetic approaches; therefore, in this chapter we will mainly focus on the key structural features of the recently proposed peptidic organocatalysts, describing their interactions with the reacting substrates. Peptides [52] characterized by proline, such as *N*-terminal amino acid, are typically used to catalyze **asymmetric aldol reactions** more efficiently than proline. However, the peculiar arrangement of the key functional groups (the *N*-terminal secondary amine and the carboxylic acid) in the peptide plays a crucial role. This was demonstrated by Wennemers et al. in 2008 by synthesizing a series of analogues of the **tripeptide** organocatalyst H-Pro-Pro-Asp-NH_2_ **31a** (Figure 13) [53]. All of the peptides were prepared by solid phase synthesis on a 500-mg scale, and they were then isolated and used as TFA salts. It was found that small structural modifications significantly modify the conformational and catalytic behavior, making the purely rational design of peptidic catalysts very challenging. In particular, the carboxylic acid position strongly affects the enantioselectivity, whereas the carboxamide tunes the reactivity. Moreover, all of the diastereoisomers of tripeptide **31a** provided the aldol products in lower yields and stereoselectivities.

Novel tripeptides were also proposed in 2014 by Kokotos and Moutevelis-Minakakis to catalyze the asymmetric aldol reaction [54]. In this case, the organocatalysts were based on Pro-Phe (Figure 14) and used as *tert*-butyl esters in an organic solvent or water. The addition of 4-nitrobenzoic acid (20 mol%) and NaBr provided the best results in water, employing only two equivalents of ketone. The C-terminal amino acid determines the ability of the tripeptide (Pro-Phe-AA-OtBu) to perform efficiently in water or toluene.

Kohari and coworkers, in 2019, developed unnatural tripeptides as organocatalysts for the asymmetric aldol reaction of isatins with acetone [55]. The authors hypothesized that high enantioselectivity could be achieved by employing a tripeptide that engaged a peculiar conformation due to the presence of a D-amino acid. In this case, H-Pro-Gly-D-Ala-OH, characterized by the D-alanine as the C-terminal residue, provided the best enantioselectivity (up to 97% *ee*). The transition state structure was investigated via DFT calculations, which confirmed the essential role of the C-terminal D-alanine residue.

The same research group proposed the tripeptide **33**, which consists of 4-*trans*-siloxyproline as *N*-terminal residue, the central L-*tert*-leucine unit, and glycine as C-terminal amino acid (Figure 9), as an efficient organocatalyst for the asymmetric aldol reaction between acetone and α-ketoesters [56] or α-keto amides [57], as it generated optically active acyclic tertiary alcohols. The achieved enantiocontrol was higher for α-keto amides than α-ketoesters; however, 20 mol% of catalyst and five days were required to obtain good yields at −40 °C. Experimental data and DFT calculations showed that the *tert*-butyl group played a remarkable role in the reaction rate and enantioselectivity.

Not only tripeptides, but also **dipeptides** are able to promote a direct asymmetric aldol reaction. In 2009, Chandrasekhar proposed the new proline–threonine dipeptide, **34** (Figure 10), which was applied in the reaction between various aldehydes and acetone [58]. Due to the lipophilic silylated protecting group present on the threonine-OH, the best results were obtained in CHCl_3_. It was found that, in this case, the enantioselectivity was higher than that achieved using the known catalyst with methyl ester and the free hydroxyl group in threonine moiety [59].

Three enantiomerically pure *N*-(1-carbamoyl-1,1-dialkyl-methyl)-(*S*)-prolinamides, resembling dipeptides involving proline and characterized by two amide functional groups (Figure 15), were synthesized by Drabina et al. and tested as organocatalysts in the aldol reaction [60]. The results were mediocre, with the (*S*)-valinamide-derived catalyst **35c** (tertiary and not quaternary α-carbon) providing the best enantiocontrol.

Zhang and Wang, in 2015, prepared a family of dipeptide-like organocatalysts (Figure 16) that were applied with low loading (1 mol%) in the asymmetric aldol reactions in brine, which proved to be a better medium than water [61]. Aromatic aldehydes and acetone provided good yields and moderate enantioselectivities, whereas using cyclohexanone provided high yields, good *ee*s*,* and excellent *de*s. A multi-gram scale synthesis of these catalysts using commercially available materials was reported by the authors.

Aiming to develop sustainable enantioselective reactions, in 2015, Juaristi and co-workers applied three novel proline-based α,β-dipeptides to the asymmetric aldol reaction under solvent-free, high-speed ball milling (HSBM) conditions (Figure 17) [62]. The transformation proceeded in shorter times and with higher stereoselectivity with respect to the conventional protocols. The organocatalysts’ synthesis was carried out in solution, employing propanephosphonic acid anhydride (T3P) as an activating agent. Regarding the catalysts structure, it was found that: (i) the presence of a second stereocenter in the α,β-dipeptide did not improve the stereoselectivity, (ii) the free carboxylic group (**37b** vs. **37a**) had no effect on the stereoselectivity and, (iii) due to the poor solubility of **37b** under HSBM conditions, the aldol reaction showed low conversion.

In 2018, Moutevelis-Minakakis and Kokotos synthesized proline-based fluorinated dipeptides as organocatalysts for the asymmetric aldol reaction both in organic and aqueous media [63]. The catalyst design aimed to improve the hydrogen bonding interactions and/or change the catalyst conformation that was adopted in the transition state (Figure 18). It was established that both the amide and fluorine moieties are essential to obtain good results in both organic and aqueous media (brine), with the 2-trifluoromethyl-aniline **38e** being the best-performing dipeptide.

The same authors, in the same year, also proposed new hybrids which combined a dipeptide moiety with a 2-pyrrolidinone scaffold (Figure 19), derived from pyroglutamic acid, which promoted the asymmetric aldol reaction in both organic and aqueous media [64]. The authors observed: (i) a beneficial effect of the pyrrolidinone ring with respect to simple dipeptide, (ii) a matched effect between the peculiar absolute configurations of Phe and the stereocenter on the pyrrolidinone, (iii) that, in brine, more distance between the Pro-Phe dipeptide backbone and 2-pyrrolidinone is profitable, and (iv) that substituted pyrrolidinones provided worse results. Catalyst **39a** was successfully recovered (73%); however, the enantioselectivity dropped when it was reused in a second reaction. These catalysts also showed interesting catalytic activity in the aldol reaction of ketone with ketones.

Effective organocatalytic enantioselective direct aldol reactions were also promoted by **pseudopeptides**. In 2011, Pearson and coworkers investigated a series of structures (Figure 20) that were characterized by the anthranilamide moiety [65]. They obtained a good reaction performance in employing pseudotripeptides, showing a favourite conformation which derives from an intramolecular hydrogen bonding (Figure 20c), leading to a defined binding pocket (Figure 20b). The best results with various aldehydes and ketones were achieved with catalyst **40f**, due to the free carboxylic acid, the presence of a α-substituent in the C-terminal aminoacid, and the methyl group at C-6 on the aromatic ring.

Kokotos proposed, in 2012, a prolinamide-thiourea consisting of (*S*)-prolinamide, (1*S*,2*S*)-diphenylethylenediamine, and (*S*)-di-*tert*-butyl aspartate (Figure 11) as an efficient organocatalyst for the challenging asymmetric aldol reaction between ketones and perfluoroalkyl ketones. Tertiary alcohols were obtained in high yields and good enantioselectivities were obtained (up 81% *ee*) by employing this tripeptide-like catalyst at low loading (2 mol%) [66].

Proline-based peptides were also employed in the **asymmetric Michael reaction** between ketones and nitroolefins. In 2009, Tsogoeva et al. synthesized some **di- and tri-peptides** (Figure 21) and applied them in the enantioselective construction of γ-nitroketones “on water” (without organic co-solvents). Good yields and *dr* were achieved using cyclic ketones and aromatic nitroalkenes, and the *ee*s reached a maximum of 70% [67]. The authors demonstrated the key role of sodium hydroxide as an additive and the beneficial effect of the lipophilic side chains (L-Phe and L-Val) compared to the functionalized L-Tyr residue. The peptide size seems to affect the performance of this transformation “on water,” since the addition of a C-terminal phenylalanine led to worse results and simple *L*-proline was unable to promote the reaction.

Wennemers et al. devoted many efforts to the development of peptidic organocatalysts to efficiently promote the asymmetric conjugate addition of aldehydes to nitroolefins. In 2009, they identified the tripeptide H-D-Pro-Pro-Glu-NH_2_ (**43a**, Figure 22) as a very effective organocatalyst which provided excellent performance under mild conditions and in the absence of additives [35]. The D-Pro-Pro moiety is the major factor responsible for the high stereoselectivity, whereas the C-terminal amide and the spacer to the free carboxylic acid contribute to the fine-tuning of the stereoselectivity. In 2010, the authors performed kinetic studies which revealed that, in the peptide-catalyzed process, the rate-limiting steps are both the reaction of enamine with the electrophile and the hydrolysis of the resulting imine (and not the enamine formation) [68]. These findings allowed the authors to decrease the loading of tripeptide catalyst **43a** to 0.1 mol% for a broad range of substrates. Their work demonstrated the key role of mechanistic insights based on kinetic studies for the optimization of organocatalytic reactions.

Afterwards (2017), the same research group studied the impact of the amide bond conformation on the stereoselectivity of H-Pro-Pro-Xaa-NH_2_-type peptide organocatalysts in conjugate addition reactions (Figure 23) [69]. The middle Pro unit within the catalyst was replaced with analogues of varying ring sizes to generate different *trans/cis* ratios. It was found that there was a direct correlation between the *trans/cis* amide bond ratio and the enantio- and diastereo-selectivity of structurally related catalysts. H-D-Pro-Pip-Glu-NH_2_ (**43b**, Figure 23; Pip: piperidine carboxylic acid), characterized by an enhanced population of the *trans* conformer, was identified as a highly reactive and stereoselective peptidic catalyst, and was well-performing at 0.05 mol%, which was the lowest catalyst loading achieved thus far for organocatalyzed reactions that rely on an enamine-based mechanism.

In 2012, Wennemers et al. focused on poorly reactive α,β-disubstituted nitroolefins, which are interesting due to providing synthetically useful γ-nitroaldehydes with three adjacent stereocenters (Figure 12) [70]. Fifteen tripeptides Pro-Pro-Xaa (Xaa = acidic amino acid) were evaluated and H-Pro-Pro-D-Gln-OH and H-Pro-Pro-Asn-OH proved to be efficient organocatalysts in this transformation. The combination of rigidity (H-Pro-Pro) and a certain degree of conformational flexibility (Gln-OH or Asn-OH) seemed to be the key to their catalytic efficiency. Moreover, these organocatalysts strongly favour the conjugate addition reaction over the competing homo-aldol reaction, and this high chemoselectivity becomes crucial when poorly reactive α,β-disubstituted nitroolefins are used as substrates. Mechanistic investigations showed that the configuration of all three of the stereocenters in the product was induced by the peptidic catalyst.

Shortly after (2013), the same authors also studied the asymmetric conjugate addition of aldehydes to β,β-disubstituted nitroalkenes, generating an all-carbon quaternary stereocenter adjacent to a tertiary stereocenter (Figure 13) [71]. In this case, the carboxylic acid at the C-terminus of the peptide had a detrimental effect, whereas the highest conversions were obtained with peptides bearing a methylester and/or aromatic residues. Dipeptide H-D-Pro-Pro-NHCH(Ph)CH_2_-4-Me-C_6_H_4_ (**43d**) provided the best reaction performance.

In 2016, Xie and Wang synthesised a family of novel bifunctional L-prolinamides which resembled dipeptides, characterized by the (*S*)-proline, a central aminoacidic unit and a terminal bulky amide (Figure 24) [72]. The nitro-Michael reaction promoted by catalyst **44b** (5 mol%) gave excellent results under mild conditions. It is noteworthy that α,α-disubstituted aldehydes also showed a good reaction performance.

In the same year, Deschamp and Lecouvey proposed a small library of novel bifunctional tripeptides which carry a phosphonic acid (Figure 14) [36]. The authors aimed to combine aminocatalysis and phosphonic acid activation to promote the stereoselective Michael addition of aldehydes to nitroalkenes. Their strategic synthesis relies on introducing the phosphonic acid moiety on a commercially available serine before its coupling with the dipeptide. The aspartic acid analogue **I-20a** was obtained through a Mitsunobu reaction leading to a γ-lactone, readily substituted by trimethyl phosphite, and final Boc-deprotection. The glutamic acid analogue **I-20b** was prepared starting from Garner’s aldehyde **I-19**, which was subjected to a sequence of high yielding reactions: (i) Horner–Wadsworth–Emmons reaction, (ii) double bond hydrogenation, (iii) ketal acidic deprotection, (iv) alcohol oxidation to the corresponding carboxylic acid, (v) Boc-deprotection, and (vi) esterification. After the coupling with the dipeptide, the *N*-Boc-proline and the phosphonate diester protecting groups were simultaneously removed with bromotrimethylsilane, the pH was adjusted, and the catalysts were lyophilized as sodium salts. This method avoided the use of an external base, allowing easier catalyst recovery. In fact, due to their water solubility, the catalysts were easily recyclable and could be reused over several cycles without any significant loss of selectivity. Applied in the asymmetric Michael reaction, these catalysts provided opposite absolute configurations compared to Wennemer’s catalysts. The different spatial geometry and activation mode of carboxylic acid and phosphonic acid could probably explain this divergent behavior [73]. Catalyst **45d**, derived from pre-adjusted pH to 11.5, was the most effective catalyst at a low loading (1 mol%) and used at 0 °C. It was observed that the catalysts protonation state played a remarkable role in the mechanism and reaction outcome.

In 2017, Martín’s research group was inspired by the idea to combine two chiral building blocks (amino acids and carbohydrates), and they proposed a series of new hybrid dipeptide-like organocatalysts which are structurally based on proline and pyranoid ε- or ζ-amino acids (Figure 25) [74]. The asymmetric Michael addition of aldehydes to β-nitrostyrenes provided good results under mild conditions, using the trifluoroacetic acid (TFA) salts of the dipeptides (5 mol%) and *N*-methylmorpholine (NMM). In these bifunctional organocatalysts, proline residue which was attached near the tetrahydropyran ring led to better stereocontrol, whereas the tetrahydropyran moiety (decorated with a methoxy group at C-4 in equatorial position) induced a well-defined conformation which is responsible for the catalyst’s efficiency. The best results were collected with catalyst **46k**, which involved the carbohydrate motif embedded in a ζ-amino acid.

Very recently (2020), Wennemers et al. developed a novel tripeptide which is able to catalyse the *anti*-selective conjugate addition reaction between aldehydes and nitroolefins [75]. It was a very remarkable achievement, since the numerous already-proposed amine organocatalysts promote this transformation resulting in *syn*-configured products. The crucial role in obtaining *anti* selectivity was played by the substituents that were installed at C-δ of the catalytically active proline ring, which favour the formation of the intermediate s-*cis* enamine, reacting in a Curtin–Hammett scenario more promptly than the s-*trans* isomer (Figure 15a). After a careful optimization of both its catalyst structure and reaction conditions, tripeptide **43e** (Figure 15b) was able to provide excellent yields and stereoselectivities on a wide range of substrates.

Peptides were successfully employed in the asymmetric conjugate addition of aldehydes not only to nitroolefins, but also to maleimide, generating products which were readily derivatized to pyrrolidines, lactams, lactones, and peptide-like compounds [76]. Extensive studies on the optimal catalyst (H-D-Pro-Pro-Asn-NH_2_) showed the importance of the hydrogen bonding interactions between unprotected maleimide and peptidic organocatalyst to achieve high stereoselectivity and to avoid side-reactions. These findings suggest that differential coordination, which underlies the substrate specificity of enzymes, can be implemented in peptidic organocatalysts.

### 2.3. Substituted Prolines

As has already been mentioned, proline has a leading role as a chiral organocatalyst due to its widespread application [77]. However, it also has several limitations. The design and synthesis of prolinamides (Section 2.1) and peptides (Section 2.2) represent a useful approach to overcome these issues, while maintaining the proline’s ability to form H-bonds. An alternative strategy consists of the functionalization of other portions of the proline scaffold, leaving the secondary amine and the carboxyl function unchanged.

An interesting contribution in this field was provided by Loh et al. in 2010, which reported on the rational design and synthesis of a new structurally rigid tricyclic amphibian chiral organocatalyst based on the hexahydropyrrolo[2,3-*b*]indole skeleton (Figure 16) [78]. The authors referred to it as a *chemzyme* possessing a well-organized chiral pocket which provided asymmetric induction, and a hydrophobic pocket enabling the organocatalytic reactions to proceed both in organic solvents and in water. The tricyclic catalyst **47a** was easily synthesized in four steps (esterification, acid-mediated ring closure, carbamylation, and final protecting groups removal; Figure 16a) from cheap and commercially available Nα-carbobenzyloxy-L-tryptophan in 74% total yield. Catalyst **47a** was tested in the enantioselective Michael addition of aldehydes to nitroalkenes, furnishing excellent performance unlike other catalysts such as proline, **47b**, or **47c** (Figure 16c). These findings supported the key role of this peculiar skeleton and suggested that the proline skeleton and not the hydrogen bonding interaction was responsible for its failure in this process. DFT calculations determined that the ethyl carbamate group induces the enamine to adopt the *syn* conformation. Being the process carried out in protic solvent (MeOH or H_2_O), the enamine *Si* face is hindered by hydrogen-bond networks; therefore, nitrostyrene will attack the less-hindered enamine *Re* face (Figure 16b).

In 2011, Lombardo and Quintavalla described the new *cis*-ion-tagged organocatalyst **48a**, which is characterized by an amide linkage between the imidazolium tag at C-4 and the proline ring (Figure 17) [79]. The ionic group installed at C-4 is supposed to increase the catalyst lipophilicity and solubility (also exploiting the hydrophobic bis(trifluoromethane)sulfonimide counterion, NTf_2_); however, at the same time, the ionic group should improve the catalyst activity by means of a stabilizing electrosteric effect. The organocatalyst was synthesized in four steps (amide synthesis, Cl-substitution by *N*-methyl imidazole, counterion exchange, and simultaneous removal of the two protecting groups; Figure 17a) starting from *cis*-4-amino-L-proline 4 in 50% total yield. Catalyst **48a** was tested in the asymmetric aldol condensation between ketones and aldehydes “in the presence” of water (1.2 equivalents), and displayed an excellent efficiency in terms of catalyst loading (2–5 mol%) and stereochemical outcome. Moreover, **48a** was bench-stable; therefore, it was more practical and profitable than the more sensitive *cis*-ester **48b** (Figure 17b) [80]. A combined experimental and computational investigation offered a rationalization for the higher reactivity of *cis*-ion-tagged proline compared to its tagged *trans* analogue and to the *cis* or *trans* tag-free organocatalysts [81]. The transition state involving *cis*-ion-tagged proline is significantly stabilized by a complex network of hydrogen bonds, π-stacking interactions involving the aldehyde aromatic ring, and π-interactions between the proline carboxyl group and the imidazole ring (Figure 17c).

The same research group, in collaboration with Gruttadauria’s group, proposed, in 2012, an innovative two-stage liquid–liquid biphasic homogeneous protocol for the asymmetric organocatalytic aldol reaction [82]. The *cis*-ion-tagged proline **48a** was dissolved in the liquid film of a 26 multi-layered ionic liquid which was covalently bonded to silica gel (catalyst loading 13.8 wt%). In the first stage, this catalytically-active material acts as a catalyst reservoir, delivering **48a** to the cyclohexanone phase, which allows the reaction to take place conveniently under homogeneous conditions. In the second stage, the ketone is removed under vacuum and the products are selectively extracted, whereas **48a** is redissolved in the supported ionic liquid film, acting as a catalyst sponge. The developed catalytic material can be easily reused several times (up to 15 cycles) with high cumulative productivity values (up to 523).

In 2014, Bhowmick et al. described a new organocatalyst which was synthesized from two inexpensive, naturally occurring substrates, 4-hydroxy-L-proline and abietic acid, in three simple steps with an overall yield of 75% (Figure 18) [83]. It was applied to the asymmetric aldol reactions in water in the presence of a sulfonic acid additive, furnishing excellent yields and stereoselectivities. The reaction was particularly fast, and the catalyst loading was reduced to 1 mol%.

## 3. Prolinol-Related Organocatalysts

As has already been demonstrated by the huge number of applications of proline as an organocatalyst, the five-membered secondary amine structure of chiral pyrrolidines proved to be a privileged structure which is able to activate the carbonyl substrates that are reacting with different partners. The importance of the pyrrolidine ring in determining the activity and stereoselectivity of the derived enamine was already highlighted in 2000 by List and Barbas in the aldol reaction between acetone and *p*NO_2_-benzaldehyde, where *N*-methyl-valine and piperidine-2-carboxylic acid failed to give acceptable conversions and azetidine-2-carboxylic acid was significantly less reactive and stereoselective than proline [28]. While proline and its derivatives (Section 2) are characterized by a carboxyl or amide group as a hydrogen bond donor directly linked at C-2, many other proposed catalysts possess prolinol-derived structures which are characterized by a methylene linker between the pyrrolidine ring and the variously substituted chain at C-2. In almost all cases, the substituent introduced in the sidechain is designed to be able to coordinate the electrophilic reaction partner, subsequently directing its approach selectively onto one of the two diastereotopic faces of the chiral enamine, either by protonation in the presence of an acid co-catalyst, or by dipole–dipole interactions. This strategy was introduced by Barbas back in 2002, by employing simple pyrrolidine amines in the presence of trifluoroacetic acid for the efficient asymmetric construction of quaternary carbon centers using organocatalysis [84,85].

In 2008, Headley and co-workers prepared a series of pyrrolidine-based chiral pyridinium ionic liquids (ILs), starting from the commercially available (*S*)-(2-aminomethyl)-1-*N*-Boc-pyrrolidine [86]. The catalysts **50a–d**, differing for the nature of the counter-anion, were efficiently prepared in two or three steps, starting from Zincke salt [87], with overall excellent yields (78–83%, Figure 19). These pyrrolidines were tested in the asymmetric Michael addition of cyclohexanones to nitrostyrenes, returning good yields and excellent selectivities. Catalyst **50b** was also recycled up to three times in the same reaction, with only a moderate decrease in performance during the third run.

In the same year, Xu and co-workers prepared 2-[(imidazolylthio)methyl]pyrrolidines (**51a–c**) as trifunctional organocatalysts for the same asymmetric Michael addition. The catalysts were prepared in three steps, starting from L-proline; unfortunately, the yield of the last step was not reported (Figure 20) [88]. Catalyst **51a** afforded good yields (83–95%) and selectivities (*dr* 80:20–93:7, 83–95% *ee*) in the proposed Michael addition, using salicylic acid as the co-catalyst and *i*PrOH as the solvent.

Starting from *N*-Boc-protected proline azide, which was in turn derived from *N*-Boc-L-proline [89], Wang and coworkers prepared recyclable ionic-liquid-supported pyrrolidine catalysts by exploiting a copper-catalyzed Huisgen 1,3-dipolar cycloaddition (Figure 21) [90]. Using catalyst **52a** in EtOH and in the presence of TFA as a co-catalyst, the benchmark Michael addition of cyclohexanone to nitrostyrenes afforded the desired products in high yields (85–95%) and with excellent diastereoselectivities (>99:1) and enantioselectivities (91–>99%). Furthermore, the catalyst was recovered and recycled at least eight times, with only a modest loss of its catalytic activity, and with excellent stereoselectivities.

Similarly, ionic-liquid supported pyrrolidine catalysts were prepared by Headley, Ni, and coworkers, starting from the commercially available (*S*)-(2-aminomethyl)-1-*N*-Boc-pyrrolidine and exploiting a sulfonamide linkage. Catalysts **53a,b** were thus prepared in 54 and 66% overall yield, in a short three-step synthetic sequence (Figure 22) [92]. Both catalysts, **53a,b**, were effective in the asymmetric Michael addition of aldehydes to nitrostyrenes, giving high yields and diastereoselectivities, and good enantioselectivities (up to 85%). On the other hand, **53a** was found to be more stereoselective in the case of α,α-branched aldehydes, while **53b** was more stereoselective in the case of α-monosubstituted ones. Finally, these catalysts were recycled and reused up to five times without a significant loss of catalytic activities and enantioselectivities.

A year later, the same authors proposed a new recyclable ionic liquid-supported pyrrolidine sulfonamide organocatalyst (**54**, Figure 23) for the asymmetric Michael addition of ketones to nitroolefins with high enantio- and diastereo-selectivities, which introduced the sulfonyl group on the C-2 position of the imidazolium cation. This structural modification makes the NH group more acidic and enhances the formation of hydrogen bonds that are pivotal in the transitions states of the corresponding reactions [93].

In the same year, Kilburn and coworkers prepared a series of bifunctional pyrrolidine catalysts to promote the asymmetric Michael addition of cyclohexanone to *trans*-β-nitrostyrene, with excellent stereocontrol. Among the different catalysts that were prepared, **55a,b**, incorporating two thiourea moieties in the sidechain, gave the best overall performances. These catalysts were prepared in a five-step synthetic sequence starting from commercially available *N*-Boc-L-proline and in 32 and 19% overall yield, respectively (Figure 24) [94].

During that year, Zhong et al. rationally designed a highly stereoselective pyrrolidine organocatalyst bearing a polar P=O bond on the sidechain to improve the asymmetric Michael addition of cyclic ketones to nitroolefins. The pivotal role of the polarized P=O bond in determining, by hydrogen bonding, the high stereoselectivity in the reaction (98:2 to >99:1 *dr*, 92% to >99% *ee*) was demonstrated by preparing the analogue catalysts which were lacking the polarized bond or which had a weaker polar bond (P=S), which afforded much worse results. Catalyst **56** was prepared in a four-step sequence and in 79% overall yield, starting from the commercially available *N*-Boc-L-prolinol (Figure 25) [95].

In 2009, Chen’s group prepared new pyrrolidine-camphor organocatalysts for the challenging asymmetric Michael additions of α,α-disubstituted aliphatic aldehydes [96] and of simpler aldehydes [97] to β-nitroalkenes. Just a few months later, the same authors applied similar pyrrolidine-camphor organocatalysts to the direct α-amination of aldehydes with azodicarboxylates [98]. The next year, Chen and coworkers optimized the catalyst structure by varying the substitution pattern both on the pyrrolidine ring and on the camphor scaffold, and applied the new catalysts to the conjugate addition of ketones to alkylidene malonates [99] and to the asymmetric Michael addition of aldehydes and ketones to β-nitroalkenes [100]. Among all of the catalysts that they tested, pyrrolidines **57a,b**, obtained by joining the *trans*-4-hydroxy-L-proline scaffold and camphor by a sulfide link, were the most efficient. The catalysts were prepared starting from a *N*-Boc-*O*-tosyl prolinol derivative; however, unfortunately, no details are given for their synthesis of the advanced intermediates. The starting materials can be prepared starting from the commercially available *N*-Boc-*trans*-4-hydroxy-L-proline and camphorsulfonic acid, following the reported procedures [101,102]. Thus, **57a** could be synthesized in a six-step sequence in 28% overall yield, and **57b** in a related seven-step sequence in 38% overall yield (Figure 26).

An alternative synthesis of a pyrrolidinyl–camphor bifunctional catalyst (**58**) was later proposed by the same authors, which was able to catalyse, in neat conditions, the asymmetric Michael addition of aldehydes and ketones to different nitroolefins in high yields and stereoselectivity [103]. The required key amino ketone intermediate can be prepared from camphorsulfonic acid, or eventually from the commercially available, although quite expensive, ketopinic acid. Again, the pyrrolidine-aldehyde required for this procedure can be prepared from the commercially available *N*-Boc-L-proline methyl ester. Thus, **58** was prepared in seven steps, and 11% overall yield (Figure 27).

Later that year, Wang and coworkers proposed simple pyrrolidines bearing a pyridine ring on the sidechain as catalysts for the highly enantioselective Michael addition of aldehydes and ketones to nitroolefins. The catalysts **59a**–**d** were prepared in two steps, starting from the commercially available *N*-Boc-L-prolinol (Figure 28). Detailed yields were reported for the preparation of **59a** (65% overall yield); however, only the yields of the last step were reported for **59b**–**d** [104].

Peng and coworkers prepared 4-trifluoromethanesulfonamidyl prolinol *tert*-butyldiphenylsilyl ether **60** as a bifunctional organocatalyst for the Michael addition of ketones and aldehydes to nitroolefins. The catalyst was prepared in a 12-step synthetic sequence starting from 4-*trans*-L-hydroxyproline and in 27% overall yield (Figure 29) [105]. Probably due to a typo, the authors reported in the experimental section the use of NaN_3_ in DCM instead of DMF for step (i): these conditions should be avoided, since the very dangerous diazidomethane could be produced [106]. Catalyst **60** was more recently employed by the same authors in an asymmetric Michael reaction for the preparation of chiral nonnatural 2,4-disubstituted pyrrolidines from aldehydes and nitroolefins [107].

In 2010, Du and coworkers prepared a new L-proline-based binaphthyl sulfonimide, starting from the commercially available (*S*)-2-amino-1-*N*-Boc-pyrrolidine. The catalyst **61** was prepared in 16% overall yield using a six-step synthetic sequence, and was used to profitably catalyze the asymmetric Michael addition of ketones to nitroalkenes (Figure 30) [108].

In the same year, Wang, Zhang, and coworkers prepared a sugar-based pyrrolidine as a highly enantioselective organocatalyst for the same asymmetric Michael addition, which afforeded the desired products in good yields and excellent stereoselectivities under solvent-free conditions. The catalyst **62** was prepared in 40% overall yield and using a five-step sequence starting from the commercially available *N*-Fmoc-L-prolinol, exploiting click-chemistry to insert the commercially available diacetone-D-glucose on the catalyst side-chain (Figure 31) [109].

A similar strategy was adopted by Chandrasekhar et al., in the same year, to introduce hydroxyphthalimide on the sidechain of pyrrolidine via an azide linker. The obtained catalyst was employed in the asymmetric Michael addition of ketones to nitroolefins, using an acid co-catalyst and water as the solvent; this catalyst returnedgood yields of the desired products, although with general lower stereoselectivities [110].

Hiraga and coworkers proposed, in 2010, different phosphonate-pyrrolidine derivatives as efficient organocatalysts for asymmetric Michael addition, with excellent yields and good selectivities. Among the catalysts that were prepared, the aminophosphonic acid monoester **63** was the most effective in terms of enantioselectivity. Further, **63** was prepared starting from L-proline, in eight steps, and in 42% overall yield (Figure 32) [111].

Sulfone-substituted pyrrolidines were prepared in the same year by Lin and coworkers as highly stereoselective organocatalysts for Michael addition “on water,” and they provided excellent results. The best catalyst, **64**, was synthesized in six steps and 49% overall yield, starting from L-proline (Figure 33) [112].

By combining two privileged scaffolds, Chen, Xiao, and coworkers designed a novel thiourea-amine bifunctional catalyst for the asymmetric conjugate addition of ketones and aldehydes to nitroalkenes, with diastereoselectivities up to 98:2 and enantioselectivities up to 96%. The most efficient catalyst, **65**, was prepared starting from the commercially available (*S*)-(2-aminomethyl)-1-*N*-Boc-pyrrolidine and from the cinchonidine-derived amine, in five steps and 22% overall yield (Figure 34) [113].

Similar bifunctional thiourea-pyrrolidines, derived from the reaction of 3,5-bis(trifluoromethyl)phenyl isocyanate and (*S*)-2-amino-1-*N*-Boc-pyrrolidine, were prepared in the same year by Li, Wang, Tang, and coworkers. Unexpectedly, the catalysts that were able to participate with a single hydrogen bond in the transition state of the addition were superior to the one possessing the unprotected thiourea moiety. DFT calculations were made to rationalize this peculiar behavior [114].

A bifunctional catalyst, possessing a 2-amino-pyridine ring on the sidechain of pyrrolidine, was designed in 2011 by Yu and coworkers. Catalyst **66** was prepared in five steps and 43% overall yield starting from L-proline (Figure 35), and it was able to efficiently promote the asymmetric Michael addition of ketones to nitroolefins with excellent yields and stereoselectivities in ethanol and in the presence of benzoic acid as the co-catalyst [115].

In the same year, Zhang, Zheng, and coworkers prepared a bifunctional pyrrolidine organocatalyst which possessed a 1H-benzo[d]imidazole moiety on the side-chain, which was successfully applied to the aqueous phase asymmetric Michael reaction of cyclohexanone with various nitroolefins and returned high stereoselectivities [116]. Catalyst **67** was prepared starting from (*S*)-(2-aminomethyl)-1-*N*-benzyl-pyrrolidine; however, no details were given for the synthesis of this starting material. A possible preparation starts from the commercially available L-proline methyl ester hydrochloride. Using the reported procedures, **67** could thus be obtained in six steps and 12% overall yield (Figure 36).

From 2009 to 2012, Peng and coworkers designed and synthesized a series of pyrrolidine-based organocatalysts which are characterized by various H-bond donors at the 4-position and a stereocontrolling silyl ether group in the side-chain at the 2-position (Figure 37b) [83,105,117]. The synthetic strategy consists of many steps (Figure 37a), starting from *trans*-4-hydroxy-L-proline: protection of both carboxyl and amine functions, Mitsunobu reaction and hydrolysis to invert the C4-stereocenter, re-esterification of the carboxyl group, OH mesylation, reduction to prolinol and its silylation, azidation with inversion of the C4 configuration, reduction to amine and installation of the proper H-bond donor, and final Boc removal. Catalyst **68a**, owing an aromatic thiourea, performed better than the analogues with aliphatic thiourea, sulfonamide, and squaramide at C4, and it was identified as the most efficient organocatalyst for various types of *anti*-Mannich reactions. It is noteworthy that challenging substrates in the Mannich reaction, such as sulfones with ortho-substituents or strong electron withdrawing groups on the aromatic ring, also performed well. The authors suggested that the bulky group (–CH_2_OTBDPS) shields the enamine *Re*-face, and both the thiourea protons in the catalyst form hydrogen bonding interactions with the imine nitrogen, activating the imine substrate (Figure 37c).

In 2010, Cai and Zhang designed and synthesised a novel recyclable fluorous (*S*)-pyrrolidine–thiourea bifunctional organocatalyst (Figure 38) [118]. The fluorous tag has a dual purpose: (i) its strong electron-withdrawing effect should enhance the NH acidity, thus providing a stronger hydrogen-bonding interaction with the electrophile, and (ii) it allows for the catalyst recovery (87%) by fluorous solid-phase extraction while ensuring the advantages of a homogeneous organocatalytic reaction. Catalyst **69** was prepared via the condensation of 4-(perfluorooctyl)aniline with phenyl chlorothioformate, followed by the phenol substitution by (*S*)-*tert*-butyl 2-(aminomethyl)pyrrolidine-1-carboxylate, and the subsequent Boc removal (Figure 38). It was employed in the enantioselective α-chlorination of aldehydes, and showed high activity and enantioselectivity.

In 2011, Kokotos’s group presented novel pyrrolidine-based bifunctional organocatalysts which incorporated a thiohydantoin or a 2-thioxotetrahydropyrimidin-4-one ring that was able to provide hydrogen bonds and/or a bulky environment (Figure 39) [119]. The synthesis started from the commercially available (*S*)-β-phenylalanine methyl ester which was converted to the corresponding isothiocyanate and coupled with (*S*)-*tert*-butyl-2-(aminomethyl)pyrrolidine-1-carboxylate, affording the corresponding thiourea (Figure 39a). The following acidic cyclization furnished the heterocyclic ring with simultaneous Boc removal, providing catalyst **70a**. The synthesis of the corresponding five-membered thiohydantoin derivative **71** was carried out in a similar fashion, starting from the proper α-aminoacid. The reactivity of these catalysts was evaluated in the Michael reaction between six-membered cyclic ketones and nitroolefins, in the presence of a low catalyst loading (1–2.5 mol%) and a slight excess of ketone (from 1.2 to 2 equivalents). The best performance was obtained with the six-membered catalyst **70a**; however, the substituted thiohydantoin derivatives (**71b–d**) provided similar results, suggesting that similar conformations in the transition state were adopted by the two different rings. A similar *ee* and *dr* were recorded regardless of the substituent and the absolute configuration of the thiohydantoin stereocenter. As expected for this type of catalyst, the authors postulated that the high enantioselectivity derives from stabilizing H-bond interactions between the heterocyclic ring and the nitroolefin (Figure 39b).

In the same year, Chen and Xiao investigated the same Michael addition promoted by structurally different bifunctional pyrrolidinyl-sulfamide organocatalysts (Figure 40), designed because they believed the sulfamides should possess more acidic N–H bonds than the corresponding (thio)ureas [120]. The substituted sulfamide moiety was linked to the pyrrolidine scaffold of (*S*)-*tert*-butyl-2-(aminomethyl)pyrrolidine-1-carboxylate, exploiting two similar synthetic pathways, both starting from catechol sulfate and characterized by a reversed order of the synthetic steps (Figure 40). The aryl substituted sulfamides **73** proved to be slightly superior catalysts; however, they required high catalyst loading (10 mol%), a low temperature (−10 °C), and a long reaction time to provide the best performance. It is noteworthy that nitrodienes also reacted smoothly.

In 2011, Li’s research group proposed the use of catalysts composed of polyoxometalate (POM) acid (H_3_PW_12_O_40_) and chiral diamines with different alkyl chain lengths (Figure 26) in the asymmetric direct cross-aldol reaction of aliphatic aldehydes with aromatic aldehydes in emulsion media [121]. In fact, catalysis in emulsion has attracted much attention as a strategy to improve the mass-diffusion limitation in liquid multiphase systems. This type of catalyst acts as a surfactant and the authors hypothesized that the observed high reactivity and enantioselectivity were mainly due to a metastable emulsion formed in the organic-aqueous biphasic systems. The mixture containing catalysts **74c** or **74d**, having a longer alkyl chain, formed more stable emulsions and showed higher reactivities. Catalyst **74d** can be precipitated by adding methanol and recycled three times without a significant drop in activity and selectivity.

The following year (2012), Jørgensen and coworkers described a new bifunctional squaramide-based aminocatalyst which was designed on the basis of the simultaneous dual activation of aldehydes and nitroolefins by amino- and hydrogen-bonding catalysis, respectively [122]. In particular, appropriately positioned functional groups capable of H-bonding interactions could control both the regio- and stereo-selectivity of the processes involving the remote γ-functionalization of α,β-unsaturated aldehydes. Computational studies suggested the formation of three H-bonds between squaramide and the nitro group, and a π-stacking between the nitroolefin and the dienamine (Figure 41a). The new catalysts were prepared in high overall yield (81–91%) in two steps, starting from (*S*)-*tert*-butyl 2-(aminomethyl)pyrrolidine-1-carboxylate and the proper methoxycyclobutene-1,2-dione (Figure 41b). The same authors successfully employed the squaramide-based catalyst **75a** in remarkably low loading and also in the asymmetric Diels–Alder reaction of anthracenes using nitroalkenes as dienophiles (Figure 41c) [123].

In the same year, Singh et al. described the chiral secondary–secondary diamine **77** as an organocatalyst obtained from L-proline and (*R*)-α-methylbenzyl amine in four simple and high yielding steps: Boc protection, amide synthesis, Boc removal, and amide to amine reduction (Figure 42) [124]. The catalyst design was based on the identified trend in enantioselectivity and reactivity versus the catalyst side-chain pKa value [115], revealing that a higher pKa favours higher enantioselectivity and reactivity (Figure 42). This catalyst proved to be effective in the asymmetric Michael addition of cyclohexanone to β-nitrostyrenes, although it was used in high loading (20 mol%). The absolute configuration of the stereocenter on the methyl benzylamine moiety plays no significant role.

Zlotin’s group synthesized, in 2013, a family of stereoisomeric natural pinane-derived bifunctional pyrrolidine organocatalysts (Figure 43) whose catalytic behaviour was examined in the asymmetric conjugate addition of six-membered cyclic ketones to nitrostyrenes [125]. The use of a properly positioned hydroxyl group as a hydrogen bonding donor is supposed to coordinate the nitroalkene, contributing to enhancing the stereocontrol. The 2-hydroxy-3-[(2-pyrrolidinyl)methylamino] pinanes **78a–d** with different configurations at the C-2 and C-3 stereocenters were obtained by the reduction of the corresponding hydroxyamides (Figure 43). The best stereochemical arrangement was found for catalyst **78b**, which furnished the desired products with excellent yield and *dr*, and good *ee*.

The same model reaction was selected by Singh and coworkers to test the catalytic activity of (2*S*)-2-[(phenylsulfinyl)methyl]pyrrolidine bearing a sulfoxide moiety (Figure 44) [126]. It was prepared by the reduction of Boc-L-proline to prolinol, which was tosylated and substituted with benzenethiol. The following thioether oxidation and Boc-deprotection furnished the desired catalyst in 57% overall yield as an unseparable 57:43 mixture of two diastereoisomers. It was used to synthesize γ-nitrocarbonyl compounds in excellent yield and stereoselectivity without additives, although a high catalyst loading (20 mol%) was employed. These findings confirmed that the sulfur atom stereochemistry does not affect the Michael reaction stereoselectivity. Moreover, the authors proved that the performance of the corresponding thioether and sulfone was inferior.

In 2014, Miura et al. proposed the pyrrolidine organocatalyst **80a**, characterized by the diaminomethylenemalononitrile moiety as a novel hydrogen bonding donor, to promote the asymmetric conjugate addition of cyclohexanone to nitroalkenes under solvent-free conditions [127]. The catalyst was prepared in three steps: 3,5-bis(trifluoromethyl)benzylamine and malononitrile derivative **I-22** was reacted to produce intermediate **I-23**, which was coupled with *N*-Boc-2-(aminomethyl)pyrrolidine and then deprotected (33% overall yield, Figure 45). γ-Nitroketones were obtained with good performance, although five equivalents of ketone, and 10 mol% of organocatalyst and acid additive were required. Afterwards, the same authors developed the fluorous analogue diaminomethylenemalononitrile organocatalyst **80b**, with a slight improvement in the reaction performance [128].

In 2014, Liu’s group designed a novel fluorine containing pyrrolidine organocatalyst, which was based on the remarkable impact on the pyrrolidine conformation exerted by different substituents and, in particular, by the fluorine–ammonium ion gauche effect [129]. The catalyst synthesis was performed in six steps starting from *trans*-4-hydroxy-L-proline: protection of both amine and carboxyl groups, DAST-mediated fluorination at C4, ester reduction to aldehyde and reductive amination to insert the second pyrrolidine unit, and final Boc-removal (Figure 46). To test the catalyst (10 mol%), the usual Michael addition of ketones to nitroolefins was carried out under neat conditions with an excess of ketone. It was observed that the fluorine substituent improved yield and *ee*, probably because the fluorine–ammonium ion gauche effect further stabilized the (*E*)-*trans*-C^γ^-endo favoured enamine conformation (Figure 46).

The same fluorine gauche effect was exploited by Kokotos and coworkers to make the pyrrolidine ring more rigid in the development of new pyrrolidinine-thioxotetrahydropyrimidinone-based organocatalysts [130]. Both the *trans*- and *cis*-diastereomers, bearing either a fluorine or a hydroxyl group, were synthesized (Figure 47) starting from protected 4-hydroxy-L-proline. The basic synthetic sequence to construct *cis*-catalyst **70c** consists of OH-replacement by fluorine, ester reduction and mesylation, azide insertion and its reduction to amine, thiourea synthesis and cyclization, and Boc removal. The corresponding *trans* isomer **70b** was achieved with the same pathway, preceded by the mesylation/benzoylation/hydrolysis sequence which inverted the absolute configuration at C4. The 4-hydroxy organocatalysts were obtained in the same manner after silylation of the 4-OH. The catalytic activity of the new pyrrolidines was evaluated in some asymmetric transformations, which were efficiently promoted by **70c** with an almost stoichiometric amount of reagents in brine, whereas the previous catalyst, lacking the fluorine, only worked in organic media. Both the *cis*-catalysts represent the matched case, while the *trans* ones proved to be the mismatched case.

In 2014, sugar amide-pyrrolidine derivatives owning D-glucose in its furanose form (Figure 48) were developed by Kumar et al. as organocatalysts for the asymmetric Michael addition of ketones to nitroolefins under solvent- and additive-free conditions [131]. The catalysts were easily obtained in high yields by coupling the *N*-Cbz-2-(aminomethyl)pyrrolidine with the carboxyl acid of the proper carbohydrate, followed by Cbz-removal (Figure 48). The sugar moiety was itnended to provide a bulky environment and an additional hydrogen bonding site. The hydroxy-catalyst **82b** promoted the reaction with good results, whereas the amino-catalyst **82a** furnished low enantiocontrol, probably due to the enamine formation by the primary amine.

Within a few years, the same research group proposed other structurally different catalysts. In 2014, they synthesized pyrrolidine-oxyimides from *N*-Boc-L-prolinol and hydroxyimides (Figure 49) [132]. In promoting asymmetric Michael addition in water, catalyst **83a** (10 mol%) was found to be superior to catalyst **83b** in all respects. Water is supposed to provide effective hydrogen bonding interactions, which bring the substrates closer in an appropriate orientation, leading to high stereoselectivity (Figure 49). The same organocatalysts (15 mol%) were employed in the enantioselective Michael addition of α,α-disubstituted aldehydes to nitroolefins under neat conditions at 0 °C [133]. In a possible transition state, the phthalimide could act as a steric controller and also provide H-bonding interactions involving the acid additive, generating a pocket-like compact transition state (Figure 49).

In 2016, a pyrrolidine-oxytriazole catalyst was developed for the enantioselective additive-free Michael addition of cyclohexanone to nitroolefins in water [134]. The simple synthetic protocol started from Boc-prolinol and commercially available hydroxybenzotriazole, which was coupled under Mitsunobu conditions, followed by Boc-deprotection (Figure 50).

In 2014, Zhao et al. designed and synthesized a family of new axially unfixed biaryl-based bifuctional organocatalysts (Figure 51) to be applied in the asymmetric Michael reaction of cyclohexanone with nitroolefins in water [135]. The underlying idea was that a significant hydrophobicity of the organocatalyst is required to strictly arrange the substrates and achieve high stereocontrol in Michael reactions carried out in water. Moreover, the biaryl group can direct the spatial orientation of the catalytically active functional groups. The catalysts were obtained by the reduction of the corresponding amides in low to moderate yields (Figure 51). The *C*_2_-symmetric catalysts **86** and **87** were also synthesized as comparisons. Catalysts **85a–c** showed much higher reactivities and slightly lower enantioselectivities than catalysts **86** and **87**, whereas prolinamide **88** provided a very low enantiocontrol under the same conditions.

Luo and coworkers proposed, in 2016, the simple perhydroindolinol **89** as an organocatalyst for the asymmetric Michael addition of aldehydes to nitroalkenes in brine, with excellent yields and high stereoselectivities [136]. The catalyst was prepared starting from the commercially available (2*S*,3αS,7αS)-perhydroindoline-2-carboxylic in two steps and in overall 92% yield (Figure 52).

In 2017, Juaristi and coworkers prepared different stereoisomeric pyrrolidine sulfinamides and studied them in the asymmetric Michael addition of cyclohexanone to nitrostyrenes. The best results were obtained with a combination of catalyst (1*R*,2*S*)-**90** and (*R*)-mandelic acid, giving the desired products in good yields and high stereoselectivities. The catalyst was prepared in a five-step synthetic sequence and in 43% overall yield, starting from the commercially available Fmoc-D-proline (Figure 53) [137].

Kaur’s group described, in 2018, the novel ((*S*)-pyrrolidin-2-yl)methyl-phenylcarbamate and -phenylthiocarbamate as organocatalysts for the benchmark conjugate addition of ketones to nitroolefins without additives [138]. The synthetic route consists of three simple steps, starting from *N-*Boc-L-proline: reduction to prolinol, addition to phenyl iso(thio)cyanate, and Boc removal (Figure 54). The carbamate (20 mol%) promoted the Michael addition in toluene with good results, whereas the thiocarbamate provided much worse results.

In 2019, pyrrolidine-oxadiazolone-based organocatalysts were proposed by Kundu and Pramanik, and their performances were assessed in the asymmetric Michael reaction [139]. The authors selected the oxadiazolone ring as a bioisostere for the proline carboxylic acid (similar pKa), which showed an improved solubility in organic solvents, being non-ionic. Starting from proline, three simple steps (reduction, *N*-Boc protection, and tosylation) provided O-tosyl prolinol, which was converted to cyano derivative and then to the oxime-type compound (Figure 55). The cyclization enabled by carbonyldiimidazole and the Boc removal furnished the two organocatalysts **92a** and **92b**. They were used as hydrochloride salts in the presence of a base and worked well in ethanol (**92a**) or water (**92b**).

In the same year, Mečiarová and Šebesta designed and prepared bifunctional thiosquaramide organocatalysts (**93a–c**, Figure 56) to be assessed in the asymmetric Michael addition of aldehydes to nitroalkenes, aiming to synthesize precursors of substituted chiral pyrrolidines [140]. Thiosquaramide derivatives are though to be promising catalysts, possessing higher acidity and improved solubility in non-polar solvents compared to the corresponding squaramides. Thiosquaramide catalysts were synthesized from squaric acid, which was converted into dicyclopentyl thiosquarate **I-29** (Figure 56). The sequential substitution of the two cyclopentyloxy groups with the proper primary amines led to the desired catalysts. The results obtained in the tested process for thiosquaramides were inferior to those of squaramides, especially in terms of reactivity.

The same research group proposed, in 2021, *N*-sulfinylpyrrolidine-containing ureas **95** and thioureas **94** as bifunctional organocatalysts in which the sulfinyl group could act as both an acidifying and a stereoinducing group [141]. The catalysts’ synthesis (Figure 57) started from Boc-(*S*)-prolinol, which was then converted to 2-(aminomethyl)pyrrolidine by exploiting the Mitsunobu/Staudinger sequence, which led to a more easily purified product. Isothiocyanate **I-31a** (using thiophosgene) and isocyanate **I-31b** were obtained in high yields. The following attachment of *tert*-butanesulfinamide and Boc removal provided the desired catalysts, although the addition of lithium sulfinamide to the isocyanate furnished low yields. The organocatalysts were tested in the Michael addition of aldehydes to nitroalkenes, which afforded a medium stereocontrol, with ureas performing better than thioureas and without a significant stereoinduction played by the stereogenic sulfur.

Very recently (2022), Kupai and co-workers synthesized the squaramide **77c** and thiosquaramide **93a** organocatalysts (Figure 58) [142], which were structurally very similar to those already proposed by Jørgensen [122] and Šebesta [140]. Their catalytic performances were tested in the asymmetric Diels–Alder reaction of (anthracen-9-yl)acetaldehyde and *trans*-β-nitrostyrene with good results, and were also tested in the asymmetric conjugate addition of 2-hydroxy-1,4-naphthoquinone to ethyl (*E*)-2-oxo-4-phenylbut-3-enoate (Figure 58), which provided high yields but low enantiocontrol.

In the same year, Moriyama et al. proposed a family of chiral aminomethylpyrrolidine organocatalysts (**96**) which were characterized by an aromatic sulfonamide bearing a benzylic substituent (Figure 59a) [143]. The first three steps of the synthetic strategy (tosylation, substitution with azide, and reduction) transformed (*S*)-*N*-Boc-prolinol into an aminomethylpyrrolidine intermediate. Then, the sulfonamide synthesis, its alkylation, and the final Boc deprotection led to the desired catalysts. They were applied to an enantioselective Michael/hemiaminal formation cascade reaction between α,β-unsaturated iminoindoles and aldehydes, which provided *anti*-2-hydroxy-hydro-1*H*-pyrido[2,3-*b*]indole (*anti*-α-carbolinol) derivatives (Figure 59b). This result was remarkable because the reported methods usually furnished *syn*-disubstituted tetrahydro-α-carbolinones. The best results were obtained by employing catalysts bearing both an electron deficient sulfonyl group and a benzyl substituent. DFT calculations showed that various non-covalent interactions between the enamine and the α,β-unsaturated iminoindole result in a significantly favoured transition state for the Michael reaction step, which led to high enantioselectivity (Figure 59b). The same research group also synthesized a family of iodinated organocatalysts (**97**, Figure 59c) which were assessed in the same cascade reaction with similar results [144]. The authors suggested that the iodine atom acts as a bulky functional group, although not as a halogen-bonding donor.

## 4. Diarylprolinol-Related Organocatalysts

Diarylprolinol silyl ethers, introduced independently by Jørgensen and Hayashi in 2005 [145,146], are unquestionably among the most efficient, versatile, and applied organocatalysts that have been thus far proposed [10,11,147]. Not only are they simply synthesized starting from proline, but they are also general and extremely stereoselective. Moreover, from the initial discovery of the classic enamine and iminium ion activation modes, their reactivity has been further expanded to dienamines, trienamines, tetraenamines, vinylogous, and bis-vinylogous iminium-ions [148]. In this class of organocatalysts, the diastereotopic discrimination of the different faces of the chiral reaction partner, most commonly an enamine or an iminium ion, is determined by the relevant steric hindrance of the substituent on the pyrrolidine ring, with opportunely protected diaryl-methanols being the most effective ones. Nevertheless, many different structural variations of Jørgensen–Hayashi catalysts have been proposed in the last twenty years, in particular with the aim to improve the catalysts’ chemical stability, eventually making the catalytic system recyclable, and in some cases its reactivity.

In 2008, Alexakis and coworkers prepared a series of chiral pyrrolidines which were characterized by the presence of an aminal moiety on the 2-position. Catalysts **98a–f** were prepared from commercially available Cbz-L-proline in a four-step synthetic sequence, and in moderate to good yields (37–60%, Figure 60) [149]. These catalysts were successfully tested in the Michael addition of propanal to *trans*-β-nitrostyrene, returning quantitative conversions in all cases, with **98a** exhibiting the best results in terms of stereoselectivity (75:25 *dr*, 79% *ee*). The scope of **98a** was further extended in the addition of different aliphatic aldehydes to *trans*-β-nitrostyrene and to vinylsulfone. When the catalysts were tested in the more challenging addition of cyclohexanone to *trans*-β-nitrostyrene, **98a–c** gave good conversions, while **98d–f** were almost unreactive. In this case, **98b** was found to be the best catalyst in terms of stereoselectivity (90:10 *dr*, 80% *ee*).

In the same year, Juaristi and coworkers pioneered the preparation of (*S*)-(pyrrolidin-2-yl)diphenyl methyl amine (**99**) for the enantioselective reduction of prochiral ketones which was catalyzed by the corresponding diazaborolidine with borane-dimethyl sulfide [150]. This nitrogen-substituted analogue of α,α-diphenyl-(*S*)-prolinol was prepared in six steps, starting from natural (*S*)-proline, in overall 49% yield, by a S_N_1-type reaction of the tertiary hydroxyl group using the azide anion as a nucleophile. (Figure 61).

A few years later, Zhong *et al.* prepared (*S*)-2-(azidodiphenylmethyl)pyrrolidine (**100**) through the direct reaction of an excess of sodium azide using α,α-diphenyl-(*S*)-prolinol in trifluoroacetic acid as the solvent (Figure 62). Catalyst **100** was efficiently used in the stereoselective preparation of densely functionalized *cis*-isoxazoline *N*-oxides [151]. In 2013, Lee and coworkers proposed an alternative one-pot protective group-free synthesis of **100** starting from α,α-diphenyl-(*S*)-prolinol, using a biphasic reaction media of H_2_SO_4_ and CHCl_3_. Furthermore, **99** was easily obtained directly from **100** using a Staudinger reaction (Figure 62) [152].

More recently, **100** and some of its derivatives were tested by Juaristi in the enantioselective organocatalytic Michael addition of aliphatic aldehydes to nitrostyrenes [153], and their performances were also compared with the analogue α,α-diphenyl prolinol and its derivatives [154]. In all of the organocatalytic reactions which were tested, **100** always afforded the best performances in terms of stereoselectivity, with respect to **99**.

In 2009, Maruoka explored the organocatalyzed benzoylation of aliphatic aldehydes using benzoylperoxide in the presence of different pyrrolidine catalysts and hydroquinone [155]. Among all of the organocatalysts which were tested, (*S*)-2-trityl-pyrrolidine (**101**) exhibited the best results, affording the desired product in improved yield and very good enantioselectivity. Catalyst **101** was prepared starting from the achiral pyrrolidine cyclic nitrone, in a two-step synthetic sequence, exploiting a final resolution step to obtain the enantiopure catalyst in overall 16% yield (Figure 63).

More recently, the same authors optimized the synthesis of trityl-pyrrolidines by applying the synthetic sequence to a chiral cyclic nitrone which was derived from 4-*trans*-L-hydroxyproline via TBS protection of hydroxy group and Murahashi decarboxylative oxidation. The so obtained catalysts showed comparable performances in terms of reactivity and selectivity in the asymmetric benzoylation of 3-phenylpropanal, with respect to the original catalyst **101** [156].

In the same year, Lu and coworkers explored the potential of various perhydroindole derivatives as organocatalysts in the asymmetric Michael reaction of aldehydes to nitroalkenes. In particular, silylated perhydroindolinylmethanol catalyst **102** was prepared starting from the commercially available (2*S*,3αS,7αS)-perhydroindoline-2-carboxylic acid, in a four-step synthetic sequence, and in overall 68% yield (Figure 64), obtaining excellent results in terms of stereoselectivity, even though it was slightly less reactive than Hayashi–Jørgensen organocatalysts [157].

In 2009, Headley, Ni, and coworkers prepared an analogue of Hayashi–Jørgensen silylated diphenylprolinol by using *N*-methylimidazole as the aromatic substituent. This catalyst, used in water and in combination with sodium bicarbonate, proved to be extremely efficient in the Michael addition of aldehydes to nitroolefins, returning high yields and excellent enantioselectivities [158]. The di(methylimidazole)prolinol silyl ethers **103a,b** were prepared in three steps and in 32–35% overall yield, starting from the commercially available *N*-Boc-L-proline methyl ester (Figure 65).

Later that year, Lombardo, Quintavalla *et al.* prepared an ionic-tagged analogue of Hayashi–Jørgensen organocatalysts (**104**) that displayed excellent performance in the stereoselective Michael addition of aliphatic aldehydes to nitrostyrenes. Catalyst **104** was prepared in five steps and overall 64% yield, starting from the known *N*-benzyl-α,α-diphenyl-(*S*)-prolinol (Figure 66) [159].

In 2010, Alexakis and coworkers extended the family of chiral aminal-pyrrolidine organocatalysts, starting from the commercially available *N*-Cbz-hydroxy-L-proline. Among all of the catalysts which were prepared, the ones possessing a phenoxy on the 4-position were the most efficient (**105a**,**b**). Catalysts **105a,b** were prepared in moderate yields (~25%), and by exploiting a stereospecific Mitsunobu reaction, with inversion of stereochemical configuration, to introduce the phenoxy group on the 4-position (Figure 67). Exploiting a synergistic effect, these catalysts were able to promote the α-functionalization of a wide range of linear and branched aldehydes/ketones, with excellent enantiocontrol and reactivity [160].

Almost concurrently, Bolm and coworkers [161] and Christmann, Strohmann, and coworkers [162] proposed a series of 2-silylated pyrrolidines (**106**) as analogues of 2-tritylpyrrolidine **101**. These catalysts were prepared by exploiting the stereoselective functionalization of *N*-Boc-pyrrolidine using *sec*-BuLi/(−)sparteine (Figure 68), which was pioneered by Beak, who prepared (*S*)-2-trimethylsilyl-*N*-Boc-pyrrolidine back in 1991 [163]. While the introduction of the Ph_2_MeSi group was very efficient (**106c**), Bolm reported that the Ph_3_Si group gave very disappointing results in terms of stereoselectivity (4% *ee*). Christmann and Strohmann were able to obtain this last catalyst (**106d**) by reacting, first, *N*-Boc-pyrrolidine with dimethoxydiphenylsilane, followed by substitution of the methoxy group with phenyllithium. Both **106c** and **106d** exxhibited excellent results in the Michael addition of aliphatic aldehydes to nitrostyrenes, albeit with slightly less enantioselectivities with respect to Hayashi–Jørgensen organocatalysts.

Interestingly, the increased reactivity of fluorosilanes with Grignard reagents for the synthesis of sterically hindered silanes was reported by Eaborn more than 70 years ago [164]. Exploiting the reactivity of fluorosilanes, in 2011, Franz and coworkers proposed an improved synthesis of 2-silylated pyrrolidines, both by increasing the stereoselectivity of the addition process and by further directly introducing more sterically hindered silyl substituents on the 2-position (Figure 69) [165].

In 2009, Gilmour and coworkers proposed the commercially available, although rather expensive, (*S*)-(−)-2-(fluorodiphenylmethyl)pyrrolidine (**107**) as an efficient organocatalyst for the asymmetric epoxidation of α,β-unsaturated aldehydes, exploiting the fluorine-iminium ion *gauche* effect, a conformational change caused by the charge-dipole interaction that was responsible for the preorganization of the transient intermediates involved in the secondary amine catalyzed processes [166]. The next year, the same authors published an improved five-step synthesis of **107** (Figure 70), which was characterized by the S_N_1-type replacement of the hydroxyl group of α,α-diphenyl-(*S*)-prolinol by fluorine using diethylaminosulfur trifluoride (DAST), returning the desired fluorinated catalyst in 47% overall yield [167]. In 2011, the use of catalyst **107** was successfully extended by the same authors to the challenging epoxidation of cyclic α,β-disubstituted enals, β,β-disubstituted enals, and an α,β,β-trisubstituted enal with excellent enatioselectivities [168].

The opportunity to conformationally stabilize the intermediates involved in secondary amines’ organocatalysis by fluorine insertion was similarly exploited by Alexakis and coworkers in 2011, who prepared the fluorinated analogues **108** (Figure 71) of catalysts **105** (Figure 67) and tested them successfully in different organocatalytic transformations [169]. The fluorinated catalysts **108a,b** were, again, prepared starting from the commercially available *N*-Cbz-hydroxy-L-proline and using DAST to insert the fluorine atom stereospecifically in the 4-position, which returned the desired catalysts in 41 and 44% overall yields, respectively (Figure 71).

In 2012, Cossío and coworkers exploited chiral ferrocenyl ligands for the stereoselective synthesis of unnatural densely substituted pyrrolidines **109a**,**b** (Figure 72), which possessed four defined contiguous stereogenic centers in the heterocyclic skeleton, via (3 + 2) cycloaddition reactions between azomethine ylides and *trans*-β-nitrostyrene. Interestingly, these catalysts were efficiently used to promote the highly stereocontrolled synthesis of pyrrolidine cycloadducts by employing the same chemistry which was used to prepare the catalysts themselves [170]. Later on, the same authors studied, in detail, the behavior of these organocatalysts in aldol reactions [171]; more recently, a review on the application of 1,3-dipolar cycloadditions between azomethine ylides and alkenes for the synthesis of catalysts and other biologically active compounds was published [172].

In the same year, Lombardo, Quintavalla, and coworkers prepared a new family of conformationally constrained bicyclic diarylprolinol silyl ethers (**110a–d**), starting from the commercially available *N*-Cbz-hydroxy-L-proline and exploiting an intramolecular Mitsunobu reaction to concurrently invert the chiral center on the 4-position and to assemble the lactone required for the installation of the aromatic substituents (Figure 73). Catalysts **110a–d** were successfully tested in different organocatalytic transformations, with **110d** returning stereoselectivities that were similar or greater than Hayashi–Jørgensen organocatalysts [173]. A detailed study of the reactivity of **110a–d** in organocatalytic transformations was published a few years later [174].

In 2014, Kesavan and coworkers prepared a series of bifunctional pyrrolidine-thiourea organocatalysts and tested them in the Michael addition of 1,3-dicarbonyls to nitroolefins. Catalysts **111a–c** were prepared in modest overall yields (8–11%) using a seven-step synthetic sequence, starting from *N*-trityl-prolinol, while **111d** was prepared starting from Juaristi’s azide [150,153,154] in 24% overall yield (Figure 74) [175].

In 2015, Tu and coworkers reported the synthesis of a family of spiro-pyrrolidines and their application to the catalytic asymmetric Michael addition of nitrometane to 3,3-disubstituted enals for the construction of all-carbon quaternary centers. The catalysts **112a**,**b** were prepared in 29% overall yield using an eight-step synthetic sequence in which the pyrrolidine ring was assembled by a tandem hydroamination/semipinacol rearrangement and the stereochemistry was controlled using a simple chiral auxiliary (Figure 75) [176]. Interestingly, while Hayashi–Jørgensen organocatalysts failed to promote the Michael reaction, catalyst **112b** exhibited excellent results both in terms of reactivity and selectivity. Later on, catalyst **112b** was also successfully employed in the construction of hydrophenanthridine derivatives using an Aza-Michael/Michael/Aldol cascade [177].

Similar pyrrolidinyl spirooxindoles were more recently proposed by Wang, Wan, and coworkers for enantioselective aldol condensation between isatins and acetone, with good activities (up to 97% yield) but only moderate stereoselectivities (up to 82% *ee*) [178].

In 2017, Melchiorre and coworkers prepared sterically hindered 4-bis-fluorinated pyrrolidine organocatalysts to efficiently catalyze the photochemical enantioselective β-alkylation of enals [179]. Catalyst **113**, more recently used by the same authors and others in photocatalytic transformations [180,181,182], was also prepared in seven steps and in 24% overall yield by Alemán and coworkers, starting from the commercially available *trans*-4-hydroxy-L-proline (Figure 76).

In 2018, capitalizing on previous results by Ellmann [183], Ruano [184], De Kimpe [185], and Reddy [186,187], Prasad reported an effective diastereoselective synthesis of diphenylprolinol methyl ether through the addition of lithium anions to chiral sulfinimines bearing a good leaving group on the 4-position [188]. The required diphenylmethanol methyl ether can be prepared in many different ways, a particularly attractive one being the metal-free boron trifluoride catalyzed etherification which was recently proposed by Chen, Xiong, and coworkers [189]. Organocatalyst **114** [190,191] could be obtained using this procedure with >99% *ee* in a five-step sequence and in 48% overall yield (Figure 77).

In 2019, Tu and coworkers prepared spiro bifunctional organocatalysts and applied them via a cascade Mannich/acylation/Wittig reaction to the total asymmetric synthesis of naucleofficine I and II, which are natural products isolated from *Nauclea genus* and exhibit antibacterial and antiviral biological activities. The most efficient catalyst (**115**) was obtained in four steps, starting from the accessible *N*-Boc-1-azaspiro[4.4]nonan-6-one (Figure 75) in 68% overall yield (Figure 78) [192].

## 5. Conclusions

In this review, the synthetic protocols reported in the literature from 2008 to 2022 for the synthesis of chiral pyrrolidines that are able to efficiently promote organocatalytic transformations have been summarized. More than 20 years of intense academic research from the initial proposals by List, Barbas, and MacMillan have allowed researchers to reach outstanding results; however, the field of organocatalysis is not yet exhausted. In his 2021 Nobel Prize lecture, MacMillan envisioned a future “*fuelled by sustainable catalysis*,” namely by “*organocatalysis, photocatalysis, biocatalysis and electrocatalysis*”. In the same year, List also stressed the importance of organocatalysis as an emergent technology, imaging a future in which asymmetric organocatalysis will be applied in a sustainable fashion and will “*play a major role in large-scale processes in fine chemical, pharmaceutical, and chemical industries*” [193,194]. The actual real-world applications of organocatalytic transformations are still very scarce and are limited to the use of Jørgensen–Hayashi diphenylprolinol trimethylsilyl ether. Thus, the proper design of new economic and efficient organocatalytic systems still remains an exciting challenge that will contribute to driving our future towards sustainability in fine chemical and pharmaceutical applications. This review aims to summarize the state of the art in the synthesis of chiral privileged pyrrolidine-based catalysts and hopes to inspire exciting new discoveries in the field of organocatalysis.

## Data Availability

Not applicable.

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
