# Peer review of "Recent Advances in Asymmetric Synthesis of Pyrrolidine-Based Organocatalysts and Their Application: A 15-Year Update"

_molecules, 2023, doi:10.3390/molecules28052234_

Round 1

Reviewer 1 Report

This review summarizes recent advances in the development of pyrrolidine-based organocatalysts. During the last twenty years, asymmetric organocatalysis has emerged as a very powerful tool for the syntheses of chiral molecules. The key to this success is the development of various types of novel organocatalysts. Pyrrolidine-based  organocatalysts for sure is a representative class. Overall this is a very nice review. I would suggest to publish after minor revision.

Suggestions:

Although L-proline, imidazolidinones, and Hayashi-Jørgensen catalysts might have been introduced in some reviews, the authors should still briefly introduce the structures of these representative and highly important catalysts in this paper, at least add a figure to show the structures of them. 

A recent report on Asymmetric Michael Reaction of Malononitrile and α,β-Unsaturated Aldehydes Catalyzed by Diarylprolinol Silyl Ether should be incorporated (Synlett 2022; 33(18): 1831-1836.)

Reviewer 3 Report

This is an excellent review on pyrrolidine-based organocatalysis, very comprehensive coverage of the recent development in the field. I only have some minor suggestions for revision.

-  Add a Figure in the Introduction to show the scope on pyrrolidine organocatalysis.  It will help readers to understand the important of this class of organocatalysis.

-     Also in the Introduction, add a Figure to list the representative pyrrolidine organocatalysts including 1) prolinamides, peptides, substituted prolines, prolinol-related, and diarylprolinol-related organocatalysts. It will serve as a content to help reader to know the scope of the paper at very beginning.

-     “chiral substituted pyrrolidines” appeared multiples in the text. Not sure if “substituted chiral pyrrolidines” is a better term.

-        Some format related issues:

P2, L 44, “etc.” should be italic.

Fig 14, fix the bond angle connecting to carbonyl to make it similar as Fig 13.  Similar for structures in Figs. 11&18.

Figs 24& 26 are bigger than other ones.

Inconsistent use of “NH” vs “N-H” for pyrrolidines.  Same for “NBoc” vs “N-Boc”…

Words like “Cat” “Acetone” in reaction Schemes shouldn’t be capitalized.
